# Obtaining Polyvinylpyrrolidone Fibers Using the Electroforming Method with the Inclusion of Microcrystalline High-Temperature Phosphates

**DOI:** 10.3390/ijms25042298

**Published:** 2024-02-15

**Authors:** Marina Vladimirovna Papezhuk, Sergei Nikolaevich Ivanin, Roman Pavlovich Yakupov, Vladimir Yurievich Buz’ko, Igor Vladimirovich Sukhno, Anna Nikolaevna Gneush, Iliya Sergeevich Petriev

**Affiliations:** 1Faculty of Chemistry and High Technologies, Kuban State University, 350040 Krasnodar, Russia; marina-marina322@mail.ru (M.V.P.); ivanin18071993@mail.ru (S.N.I.); yakupov@sfedu.ru (R.P.Y.); buzkonmr@mail.ru (V.Y.B.); 2Laboratory of Advanced Nanobiotechnologies, Kuban State Agricultural University, 350044 Krasnodar, Russia; sukhno_igor@mail.ru (I.V.S.); gneush.anna@yandex.ru (A.N.G.); 3Laboratory of Problems of Stable Isotope Spreading in Living Systems, Southern Scientific Centre of the Russian Academy of Sciences, 344006 Rostov-on-Don, Russia

**Keywords:** phosphates, hydroxyapatite, polyvinylpyrrolidone, electrospinning, fibers, SEM, XRD, IR spectroscopy

## Abstract

The results of the synthesis of microcrystalline calcium phosphates such as hydroxoapatite, pyrophosphate, and tricalcium phosphate are presented herein. The influence of the addition of polyvinylpyrrolidone (PVP) on the phase characteristics of the resulting high-temperature ceramic sample is considered. The X-ray results show that hydroxyapatite (HAp) consists of a Ca_5_(PO_4_)_3_(OH) phase, while the sample with the addition of polyvinylpyrrolidone contains β-Ca_3_(PO_4_)_2_ (65.5%) and β-Ca_2_P_2_O_7_ (34.5%) phases calcium phosphates (CPs). IR spectroscopy was used to characterize the compositions of the samples. An important characteristic of the obtained samples is the elemental Ca/P ratio, which was determined via energy-dispersive analysis. The data obtained are consistent with the composition of dental enamel apatites, namely, in the CPs (1.27) and HAp (1.40). SEM was used to study the morphology of the surfaces of hydroxyapatite particles. Polyvinylpyrrolidone polymer fibers were obtained using the electroforming method with the inclusion of CPs in the composition. The fibers were oriented randomly, and nanoscale hydroxyapatite particles were incorporated into the fiber structure. Solubility data of the HAp, CPs, and Fibers in a physiological solution at room temperature and human body temperature were obtained. The solubility of the resulting HAp turned out to be higher than the solubility of the CPs. In turn, the concentration of Ca^2+^ in a physiological solution of PVP composite fibers with the inclusion of CPs was lower than that in powdered CPs.

## 1. Introduction

Many companies use various calcium phosphates as the main components of implants in medicine. Issues related to methods of preparation, modification, changes in the phase composition of such compounds, and the physicochemical characteristics of materials based on calcium phosphates remain relevant. Calcium phosphate cements are typical self-hardening ceramic materials used in implant dentistry. Under the influence of environmental conditions, tricalcium phosphate is hydrolyzed to non-stoichiometric hydroxyapatite [1]. Hydroxyapatite (HAp) Ca_10_(PO_4_)_6_(OH)_2_ is the most commonly employed ceramic material in orthopedic and dental therapies. This is due to its preeminent presence within the mineral composition of tooth enamel and human bone tissue (60–70%) [1,2,3,4,5,6,7]. Hydroxyapatite exhibits high biocompatibility, a complete lack of immunogenicity, carcinogenicity, and slow degradation [8,9,10,11]. Obtained by annealing the bones of large-horned cattle, hydroxyapatite is currently used in various biomedical applications. However, this natural (biological) form of hydroxyapatite has drawbacks, including the risk of infection transmission, material rejection, and the possible accumulation of heavy metals in animal bones during their lifetimes [12]. To address these issues, chemically pure synthesized hydroxyapatite can be used. This form offers several advantages, including the satisfaction of ethical and medical considerations. Much attention is paid to synthesis methods, pH control, the use of various additives, and the temperature–time characteristics of annealing during the synthesis of HAp, as these factors impact the end product’s phase composition, structural characteristics, Ca/P ratio, and specific surface area.

Alongside hydroxyapatite, various calcium orthophosphates are used in regenerative medicine. High-temperature calcium orthophosphates and composites, such as α-, β-Ca_3_(PO_4_)_2_, Ca(H_2_PO_4_)_2_, Ca_5_(PO_4_)_3_(OH), Ca_4_P_2_O_9_, Ca_10_(PO_4_)_6_O, Ca_10_(PO_4_)_6_F_2_, and β-Ca_3_(PO_4_)_2_-hydroxyapatite, are traditionally used. The most important property of such compounds is their better solubility in water compared to hydroxyapatite; therefore, pure hydroxyapatite has limited use in the field of hard-tissue regeneration and drug delivery [13]. The solubility of a material allows one to predict its behavior in the environment of the body [14]. If the solubility of a material is less than the solubility of the mineral portion of the bone tissue, then it degrades slowly. If the solubility of a material is higher than the solubility of the corresponding type of bone, then complete resorption occurs. The solubility of calcium phosphate materials increases in the following order: fluorapatite → fluorohydroxyapatite → hydroxyapatite → carbonate hydroxyapatite → tricalcium phosphate → pyrophosphate [15]. Increasing solubility will positively affect the resorbability of calcium phosphate materials [16,17,18]. But in turn, excessively high solubility can lead to voids in the implanted material, which cannot be filled quickly enough with the body’s natural HAp. Therefore, hydroxyapatite is typically incorporated into composites with calcium orthophosphates or polymers [19,20,21,22]. In addition, such a material will be most similar to natural bone tissue [23], which is a hydroxyapatite–collagen composite.

A wide range of polymers can be utilizeed for the production of such fibers. In particular, biocompatible polymers are of great interest as they meet several criteria, including a lack of a propensity to induce local inflammatory reactions, allergies, or toxic effects. Additionally, these polymer materials can biodegrade in the presence of living tissues and biological fluids. In tissue regeneration, these polymers are especially useful, as they reduce the risk of implant rejection by the body’s immune system. Examples of such polymers include polycaprolactone (PCL), polyvinyl alcohol (PVA), polyvinylpyrrolidone (PVP), and chitosan (CHS) [23,24,25,26,27,28]. These polymers can be used as implantable scaffolds for ceramic. The removal of the scaffold from the body does not require surgery, as the polymer is absorbed and degraded by the body [29]. The inclusion of ceramic materials in the polymer matrix makes it possible to control the rates of degradation and resorption [30].

Recently, composite materials in the form of non-woven fibers with ceramic additives included in them have been of particular interest. The electrospinning method makes it possible to obtain non-woven materials with fibers ranging in size from micrometers to nanometers [31,32]. This range of fiber sizes allows for greater specific surface area and flexibility in performing surface functions than any other known form of material [33]. Due to these outstanding properties, polymer nanofibers are optimal candidates for many important applications in biomedical fields: polymer nonwoven fibers are used in drug delivery systems [16], and nonwoven fibers have a large surface area suitable for cell attachment [33]. Polyvinylpyrrolidone-based fibers are of particular interest. The main advantage of PVP is its biodegradability in physiological environments. Non-toxic and biocompatible PVP does not have a negative effect on animal organisms and has an optimal absorption rate and non-toxic decomposition products [34].

The aim of this work was to study the bioresorbability of synthesized HAp and CPs as well as the composite fibers for their further use in regenerative medicine.

## 2. Results

Figure 1 shows the results of X-ray phase analysis for samples calcined at 900 °C for 2 h.

The X-ray results show that the sample of hydroxyapatite consists of a Ca_5_(PO_4_)_3_(OH) phase (PDF 00-024-0033), while the CPs sample with the addition of polyvinylpyrrolidone contains β-Ca_3_(PO_4_)_2_ (PDF 00-009-0169) and β-Ca_2_P_2_O_7_ (PDF 00-033-0297) phases.

The crystal lattice parameters for Ca_5_(PO_4_)_3_(OH), β-Ca_3_(PO_4_)_2_ and β-Ca_2_P_2_O_7_ are given in Table 1.

Figure 2 shows the Rietveld refinement data for the X-ray diffraction data obtained for HAp (Figure 2a) in the space group P63/m (a) for CPs refined in space groups R-3c and P41, respectively (Figure 2b).

From Figure 2b, it can be gleaned that the CPs consist of 34.5% β-Ca_2_P_2_O_7_ phase and 65.5% β-Ca_3_(PO_4_)_2_ phase. The Goodness-of-Fit (GOF) parameter for the HAp sample has a value of 1.67, while that for the CP sample it is 1.79.

Figure 3 shows images of the structures of the researched compounds, i.e., Ca_5_(PO_4_)_3_(OH) (a), β-Ca_3_(PO_4_)_2_ (b), and β-Ca_2_P_2_O_7_ (c).

Hydroxyapatite (Figure 3A) crystallizes with hexagonal symmetry, corresponding to space group P6_3_/m. In the HAp crystal lattice, PO_4_^3−^ tetrahedra are connected by Ca^2+^ bridges. The OH^−^ ions are aligned along the sixfold lattice axis bounded by the Ca^2+^ and PO_4_^3−^ columns, forming the so-called “apathetic channel”. Phosphate β-Ca_3_(PO_4_)_2_ (Figure 3B) crystallizes in the polar trigonal space group R-3c. Dicalcium diphosphate (Figure 3C) consists of P_2_O_7_
^2−^ groups connected by Ca^2+^ cations. It crystallizes in the tetragonal symmetry group P41.

Figure 4 shows the TG, DTG, and DSC curves of hydroxyapatite dried at 100 °C. The HA powder was heated from room temperature to 1000 °C. In the figure, when the temperature reaches 111 °C, weight loss is visible on the TG curve. In the range of 200–350 °C, HAp powder quickly loses weight. Further heating did not lead to changes in weight. The total mass loss was 19.71%.

The analysis of the functional groups in the samples was conducted using infrared spectroscopy (Figure 5). Based on literature data [2,35,36], the main characteristic vibration frequencies of PO_4_^3−^ and OH^−^ groups were determined for the synthesized samples.

In the IR spectrum of HAp (100 °C) (Figure 5) dried at 100 °C, nitrate group (NO_3_^−^; 1315 cm^−1^) and carbonate group (CO_3_^2−^; 1416 cm^−1^ and 827 cm^−1^) bands are present.

The vibrations of the OH^-^ group in HAp (900 °C) (Figure 5) are located at 3570 cm^−1^ and 631 cm^−1^ [37,38]. The band around 962 cm^−1^ is explained by the non-degenerate symmetric vibration of the P-O bond in the phosphate group [39]. The weak peak at approximately 474 cm^−1^ indicates the doubly degenerate bending vibration of the P-O bond in the phosphate group, ν^2^. The most intense peak of phosphate vibrations at 1094 cm^−1^ can be explained by the triply degenerate asymmetric stretching vibrations of the P-O bond, ν^3^. The band at 1020 cm^−1^ is also explained by stretching vibrations of the phosphate ion. The sharp peaks at 559 cm^−1^ and 600 cm^−1^ are associated with the ν^4^ mode of the O-P-O bond. The band at 962 cm^−1^ is associated with the symmetric stretching ν^1^ mode of the P-O bond, while the two strong bands at 1033 cm^−1^ and 1094 cm^−1^ are associated with the ν^3^ mode.

The IR spectrum of the CPs (Figure 5) corresponds to the spectrum of the HAp sample without additives, calcined at 900 °C, except for additional signals at 943 cm^−1^ and 727 cm^−1^. These vibrations represent vas and vs. modes of P-O-P in the P_2_O_7_^2−^ ion, respectively [40]. Functional groups are present in the pyrophosphate phase [33]. The bands at 1213 cm^−1^ and 1157 cm^−1^ can be attributed to the stretching vibrations of the P=O ion in PO_4_^3−^ [41]. Signals from the OH group are absent in the spectrum.

The results of IR spectroscopy are consistent with the data from thermal and X-ray phase analysis.

Scanning electron microscopy (SEM) was used to study the microstructures, particle sizes, and surface morphology of samples annealed at different temperatures. Microstructure was studied in the detection-of-secondary-electrons mode. The advantage of using the secondary electron detection mode is that it grants one the ability to study surface morphology with the dependence of contrast on relief [42,43].

Figure 6 shows SEM photographs of the synthesized samples.

Based on the obtained photographs of the microstructure of the studied HAp and CPs, histograms of particle size distribution were constructed (Figure 7).

To determine the qualitative composition of the synthesized samples and confirm the absence of impurities, energy-dispersive X-ray microanalysis (EDX) was carried out using an IncaX-sight energy-dispersive detector (Oxford Instruments, Abingdon, England) installed on a JEOL JSM-7500F scanning electron microscope (JEOL, Tokyo, Japan). The X-ray spectral analysis method allows for both the qualitative and quantitative analysis of samples without destroying their integrity [44,45]. To carry out energy-dispersive analysis, the synthesized samples were pressed into tablets (Figure 8f) using a pressure of 5 tons/cm^2^. Figure 8 shows the scanning area of the sample, 30 × 30 μm (a); a map of the distribution of elements in the tablet of the HAp sample, namely, O (b), P (c), and Ca (d); and the EDA spectrum of the HAp sample (e). It is shown that the distribution of elements was uniform over the entire area of the tablet. The study of the samples’ CPs was carried out similarly. The results of elemental analysis are summarized in Table 2.

It is worth noting that the CP sample does not contain carbon atoms, which indicates that the entire polymer additive burned out during the annealing of the sample.

Figure 9 shows photographs of the CPs included in the PVP fibers. At magnifications of 2500× and 5000×, it can be seen that the fibers have a diameter of up to 2 μm. The powder of the CPs is evenly distributed throughout the volume of the fibers.

Figure 10 shows an image of a collector with composite fibers sprayed onto it.

The fibers formed a homogeneous mat (Figure 10). The absence of defects on the surfaces of the resulting polymer fibers indicates the optimal selection of parameters for producing micron fibers via electrospinning. As can be seen in Figure 9, the fibers are randomly oriented and have a smooth surface.

The resorption capacity of a material is determined using solubility values. Table 3 shows the solubility values of the samples, namely, HAp, CPs, and Fibers (CPs), in a physiological solution at room temperature (20 ± 1 °C) and human body temperature (37 ± 1 °C). The content of Ca^2+^ ions in the solution was determined using trilonometric titration in the presence of eriochrome black T with an ammonia buffer (pH 9–10) [46].

## 3. Discussion

The powders of the studied samples were synthesized in the laboratory via chemical deposition from a solution. When performing this synthesis method in an alkaline environment, it is critical to understand that, depending on the synthesis conditions, carbonate-containing hydroxyapatite is often obtained. Substitution groups can cause characteristic changes in lattice parameters, crystallinity, crystal symmetry, thermal stability, morphology and solubility, and physical, chemical, and biological characteristics. Accordingly, it should be taken into account that CO_2_ is released from the sample at a temperature of 450–950 °C, and to obtain Ca_10_(PO_4_)_6_(OH)_2_, it is necessary to carry out subsequent high-temperature annealing of the sample. We studied samples of HAp dried at 100 °C, HAp annealed at 900 °C, and CPs obtained using a similar synthesis method. The difference was the use of a polymer additive (PVP) during the synthesis.

In the X-ray diffraction pattern (Figure 1) of HAp, reflections belonging to the Ca_5_(PO_4_)_3_(OH) phase (PDF 00-024-0033) can be observed. In the X-ray diffraction pattern of the CPs, two phases can be identified, namely, calcium phosphate Ca_3_(PO_4_)_2_ (PDF 00-009-0169) and calcium pyrophosphate β-Ca_2_P_2_O_7_ (PDF 00-033-0297), which is consistent with the data [47]. To quantify the phases in the CPs, the Rietveld method [48] was used (Figure 2). We used HighScore Plus version 3.0e (3.0.5) and CIF files (Figure 3) from the following works: for Ca_5_(PO_4_)_3_(OH), [49]; for β-Ca_3_(PO_4_)_2_, [50]; and for the β-Ca_2_P_2_O_7_ phase, [51]. CPs is a two-phase powder with a phase ratio of 34.5%:65.5% for β-Ca_2_P_2_O_7_ and β-Ca_3_(PO_4_)_2_, respectively (Figure 2). Apparently, the polymer additive of PVP affects the thermal stability of hydroxyapatite. As a result of thermal treatment, calcium hydrogen phosphate (СаНРО_4_) is formed from brushite (СаНРО_4_·2H_2_O). In turn, at about 700 °C, calcium hydrogen phosphate (СаНРО_4_) converts to calcium pyrophosphate (Ca_2_P_2_O_7_). According to X-ray diffraction studies of Ca_2_P_2_O_7_, it exists in three different forms depending on the temperature, according to Equation (1) [52,53,54].
(1)$$\gamma \text{-}\mathrm{Ca}P_{2}O_{7}\;\overset{750{}^{\circ}C}{\to }\;\beta \text{-}\mathrm{Ca}P_{2}O_{7}\;\overset{1171{}^{\circ}C}{\to }\;\alpha \text{-}\mathrm{Ca}P_{2}O_{7}$$

During high-temperature sintering at around 900 °C, the formation of whitlockite occurs, according to Equation (2) [55].
Ca_10_(PO_4_)_6_(OH)_2_ + Ca_2_P_2_O_7_ = 4Ca_3_(PO_4_)_2_ + H_2_O(2)

Thus, under these synthesis conditions, high-temperature calcium orthophosphates are formed. It should be noted that for the method of synthesis used and the corresponding calcination temperature, the Ca_3_(PO_4_)_2_ phase may form as an impurity phase in both samples, which was also confirmed in [2,35].

According to the results of thermogravimetric (TG) analysis (Figure 4), the mass of hydroxyapatite (HAp (100 °C)) powder without additives decreased by 19.71% when HAp (100 °C) was heated to 948.6 °C. With further heating, the substance remained thermally stable [47,56]. The process of the thermal decomposition of hydroxyapatite samples obtained without additives and preliminarily dried at 100 °C can be divided into stages. The first is the removal of sorbed water [57,58] and carbon dioxide (1.25% 111 °C; 0.54% 162 °C), and the second is the decomposition of ammonium nitrate and hydroxyapatite carbonate with the elimination of CO_2_ (16.19% at 353 °C). The peak on the DTG (Figure 4) curve at 248 °C is associated with the decomposition of ammonium nitrate.

As expected, when considering the IR spectrum (Figure 5) of HAp dried at 100 °C, this spectrum contained peaks at 1416 cm^−1^ and 827 cm^−1^, correlated with the CO_3_^2−^ group [39]. Their presence is due to the high activity of Ca(OH)_2_ and the presence of carbon dioxide in the air. In addition, the NO_3_^−^ group is also present in the IR spectrum of HAp (100 °C). Absorption bands of the NO_3_^−^ group are also present in the IR spectrum in the regions of 1315 cm^−1^ and 827 cm^−1^. When the samples were calcined at a temperature of 250–300 °C, ammonium nitrate, which was sorbed on the surface of the sample, decomposed; accordingly, the NO_3_^−^ signal disappeared in the IR spectrum of HA annealed at 900 °C. The appearance of additional signals in the IR spectrum of an HA sample obtained with the addition of PVP may indicate the appearance of another phase. The signals at 943 and 727 cm^−1^ belong to νas P-O-P and νa P-O-P in the P_2_O_7_^2−^ ion, respectively [33,40].

Analysis of the obtained microstructure photographs (Figure 6) led us to the conclusion that the synthesized sample HAp (100 °C) consisted of agglomerates with a size of 20–30 μm (Figure 6a), consisting of smaller particles. At a magnification of 30,000×, it can be seen that the agglomerates of the hydroxyapatite sample consist of two fractions of thread-like particles: one with a diameter of about 100 nm and a length of several microns and one with a diameter of more than 10–20 nm (Figure 6b). Figure 6c,d show microstructure photographs of HAp (900 °C) obtained at magnifications of 2500× and 30,000×, respectively. It should be noted that upon calcination of the hydroxyapatite sample at 900 °C, the sintering of thread-like particles occurs, followed by the formation of a microcrystalline homogeneous powder. Figure 6e,f show photographs of the CPs. At a magnification of 30,000×, the CPs consists of agglomerates of sintered submicron-sized particles with a non-uniform shape.

The molar ratio of Ca/P in the mineral phase of bone tissue ranges from 1.37 to 1.67. [18,59,60]. In our case, the molar ratio of CPs is 1.27, and that for HAp is 1.4 (Figure 8e, Table 2). Such a ratio corresponds to the bone composition of tooth enamel (1.33–2.00) [61].

Synthetic hydroxyapatite has low solubility, and it is limitedly used as an implant material in its pure form [13,62]. Therefore, additional manipulations are required to increase the overall solubility (bioresorbability) of hydroxyapatite-based materials. The rate of dissolution of a material depends on the magnitude of its physical and chemical characteristics, such as its composition, specific surface area, and the presence of defects in the crystal lattice [63]. In practice, the bioactivity of a material is characterized by the rapidity of fusion with the bone tissue of the body through the formation of HAp on the surface of the material as a result of the hydrolysis of calcium phosphates or due to the deposition of HAp onto the surface of the material from a supersaturated intercellular fluid [64,65,66]. Data on the solubility of the samples are given in Table 3. The solubility of CPs turned out to be lower than the solubility of HAp by an order of magnitude. Coarse-crystalline ceramics based on β-Ca_3_(PO_4_)_2_ and β-Ca_2_P_2_O_7_ are resorbed more slowly, which can be attributed to the difference in the particle size of HAp and CPs. A histogram of the particle size distribution is shown in Figure 7. From the data, it can be gleaned that HAp consists mainly of particles with a size of 100–300 nm, and CPs consists of particles with a size of 200–400 nm. The Ca^2+^ concentration value (mol/L) in the physiological solution of PVP composite fibers with the inclusion of CPs was lower than that of powder CPs, which is associated with the formation of hydrogel when PVP fibers swell in a physiological solution. Thus, PVP makes it difficult for free calcium ions to escape from the surfaces of the fibers into the solution.

## 4. Materials and Methods

### 4.1. Materials

Ca(NO_3_)_2_·4H_2_O (purum), (NH_4_)_2_HPO_4_ (pro analysi), 25% aqueous ammonia solution NH_4_OH (purissimum speciale), and polyvinylpyrrolidone (PVP, Mw 40,000) were purchased from Vecton Company (Krasnodar, Russia). Standard titer EDTA-Na_2_ (0.1 N) and Eriochrome black T (pro analysi) were purchased from LenReaktiv (St. Petersburg, Russia).

### 4.2. Calcium Phosphate Synthesis

Synthesis of hydroxyapatite without additives was carried out via precipitation from solution according to the procedure described in [67,68]. The scheme of the synthesis of hydroxyapatite and hydroxyapatite with additives is shown in Figure 11.

To obtain the hydroxyapatite (hereafter referred to as HAp), a stoichiometrically required amount of calcium (II) nitrate was dissolved in a specific volume of calcium nitrate solution according to out calculations. The stoichiometric Ca/P elemental ratio should be 1.67, which corresponds to the element ratio in biogenic hydroxyapatite [69]. Next, under intensive stirring, an ammonium hydrogen phosphate solution was added, and the pH of the reagents was then adjusted to 10 using a concentrated ammonia solution (ρ = 0.903 g/mL). The solution was stirred using a magnetic stirrer for 2 h while monitoring the pH and temperature; then, the solution was left to stand for 44 h. The equation of the reaction is shown in (3):10Ca(NO_3_)_2_ + 6(NH_4_)_2_HPO_4_ + 8NH_4_OH → Ca_10_(PO_4_)_6_(OH)_2_↓ + 20NH_4_NO_3_ + 20H_2_O(3)

Then, the precipitate was filtered using a Buchner funnel, washed with hot distilled water on the filter, and dried at 100 °C for 2 h and at 250 °C for 1 h. HAp was selected for analysis during the drying stage at 100 °C. After drying, the obtained sample was annealed in a muffle furnace at 900 °C for 2 h. A modified synthesis method was also used to obtain CPs. PVP polymer was added to vary the phase composition of the material.

### 4.3. Electrospinning of Fibers

A 10% polymeric viscous solution of polyvinylpyrrolidone in ethanol was prepared. A total of 0.02 g of CPs was dissolved in 0.3 g of the polyvinylpyrrolidone solution. A schematic of the needleless electrospinning installation is shown in Figure 12.

A sample of hydroxyapatite was ground in an agate mortar. Double-distilled water was used as a solvent. The prepared solutions were used for electrospinning. In this study, we used a needleless electrospinning design that allows the formation of fibers from a solution flowing under the influence of gravity along a vertically oriented spinning electrode—a metal wire in which a 0.2 mm wire is wound on a 1 mm wire. The resulting fibers were collected on a grounded rotating cylindrical collector located 10 cm from the spinning electrode. The applied voltage was 18 kV, and the forming electrode and the collector were created by connecting to a high-voltage source. The fibers produced were collected on a rotating collector electrode consisting of eight metal rods that, when in motion, acted as a cylindrical surface. After the electrospinning process, the composite fibers were air-dried for 24 h.

### 4.4. Experimental Method

To determine the phase composition of hydroxyapatite samples, powder X-ray phase analysis was carried out on an X-ray diffractometer XRD-7000 (Shimadzu, Japan). The samples were studied in the range of 2θ angles from 20° to 60°, using Cu Kα radiation (λ = 1.5406 Å). XRD measurements were taken at room temperature.

The thermal degradation of the samples was studied using a Netzsch STA 409 PC/PG instrument in the range of 25–1000 °C in an open platinum–rhodium crucible in air using Al_2_O_3_ as an inert standard. The heating rate was 10 °C per minute.

The functional groups in the synthesized hydroxyapatite samples were determined using Fourier transform infrared spectroscopy (FTIR). FTIR spectra were obtained using a Fourier transform spectrophotometer, VERTEX 70 (BRUKER, Ettlingen, Germany), in the middle-infrared range in the frequency range of 400–4000 cm^−1^.

Measurement of the fiber size as well as analysis of the surface morphology of hydroxyapatite samples were carried out using a JEOL JSM-7500F ultra-high-resolution scanning electron microscope (JEOL, Tokyo, Japan).

A histogram of particle size distribution was constructed using the ImageJ 1.52u software product.

Elemental analysis of hydroxyapatite samples was carried out using an IncaX-sight energy-dispersive detector installed on an electron microscope.

The solubility of the samples was determined in physiological solution according to the total content of Ca^2+^ ions. The sample was kept in solution for 7 days at 20 °C and 37 °C. The solution volume and weight of the samples were determined according to the recommendations of ISO 10993-5-2023 [70]. The Ca^2+^ content in the solution was determined using complexometric titration.

## 5. Conclusions

In this work, samples based on calcium phosphates were synthesized. The calcination of the synthesized samples at a temperature of 900 °C led to the formation of hydroxyapatite and, in the case of using the PVP additive, to the formation of a two-phase powder: pyrophosphate and calcium phosphate. The presence of these two phases in the CPs sample can help improve the osteoinductive properties of the hydroxpapatite-based material by increasing the pH in the area where the material makes contact with bone apatite and promote the bioresorption of bone apatite [71,72]. The molar calcium–phosphate ratio, Ca/P, in the samples turned out to be close to the composition of tooth enamel. This indicates the promise of using these synthesized materials, HAP and CPs, in regenerative medicine and dentistry. It should be noted that PVP composite fibers with incorporated CPs particles are of independent interest for various applications since they have improved physical properties (strength, elasticity, etc.) compared to traditional composites [73]. Fiber diameter, specific surface area, pore size, and total pore volume significantly influence the diffusion of the liquid in which the nanofibers are immersed and influence the release of the bioactive substance according to the obtained solubility data in a physiological solution. The advantage of nanofiber materials is that their structure, namely, the fiber diameter, density, and thickness of the nanofiber layer, can be controlled by changing the process parameters of electrospinning [74].

## Figures and Tables

**Figure 1 ijms-25-02298-f001:**
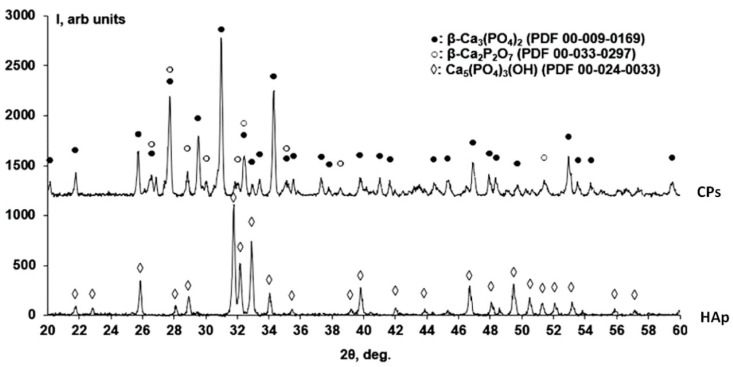
X-ray phase analysis of HAp and CPs.

**Figure 2 ijms-25-02298-f002:**
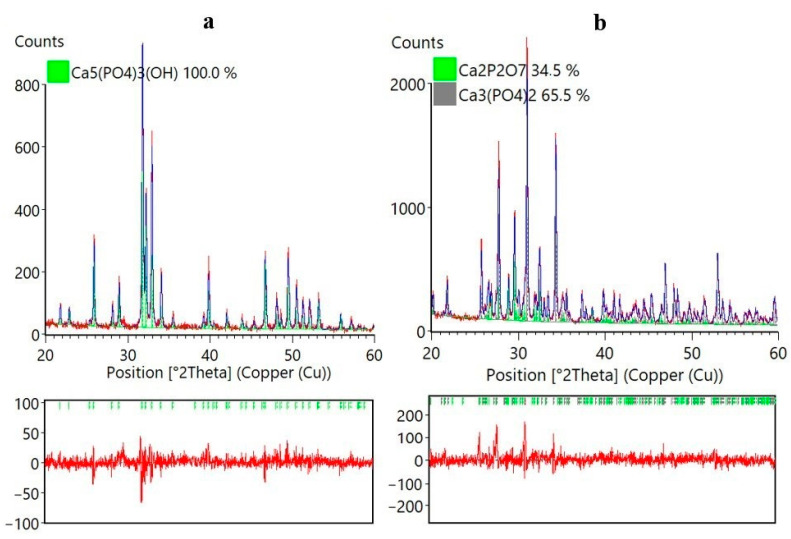
Refinement via the Rietveld method of X-ray diffraction data obtained for Ca_5_(PO_4_)_3_(OH) in the space group P6_3_/m (**a**) and for β-Ca_3_(PO_4_)_2_ and β-Ca_2_P_2_O_7_ refined in space groups R-3c and P41, respectively (**b**). The red solid line and blue solid line represent the experimental and calculated intensities, respectively, and the red line below represents the difference between them. Green marks indicate the positions of the Bragg peaks.

**Figure 3 ijms-25-02298-f003:**
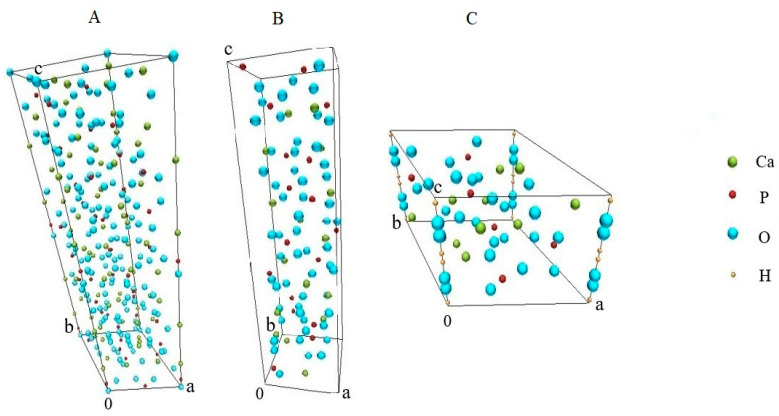
Structures of the researched compounds, according to CIF files: Ca_5_(PO_4_)_3_(OH) (**A**), β-Ca_3_(PO_4_)_2_ (**B**), and β-Ca_2_P_2_O_7_ (**C**).

**Figure 4 ijms-25-02298-f004:**
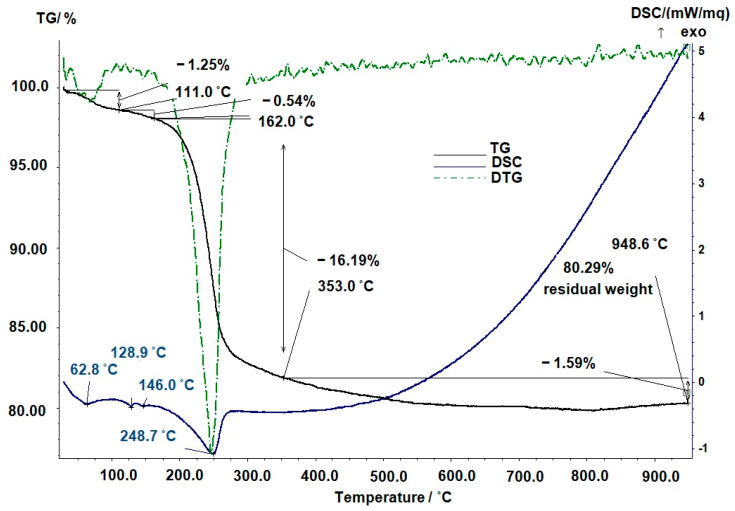
Thermogram of a hydroxyapatite sample dried at 100 °C indicating the curves: the black solid line is mass loss versus temperature (TG); the green dashed line is differential curve corresponding to the maximum rate of change in sample mass (DTG); the blue solid line is first derivative of temperature for phase transitions (DSC).

**Figure 5 ijms-25-02298-f005:**
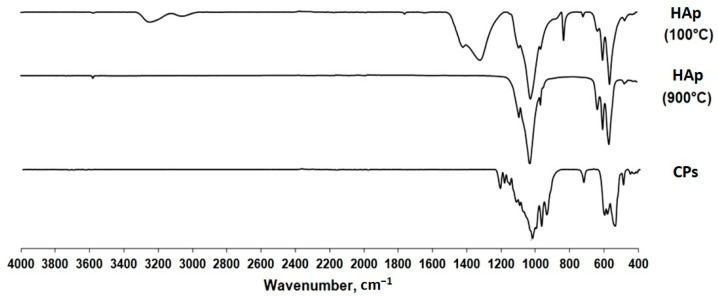
IR spectra of HAp (100 °C), HAp (900 °C), and CPs.

**Figure 6 ijms-25-02298-f006:**
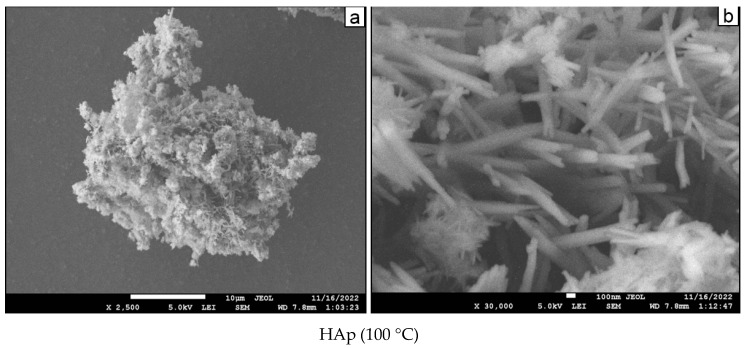
SEM photographs (2500×, 30,000×) of the microstructures of the synthesized HAp (100 °C)—(**a**,**b**); HAp (900 °C)—(**c**,**d**); CPs—(**e**,**f**).

**Figure 7 ijms-25-02298-f007:**
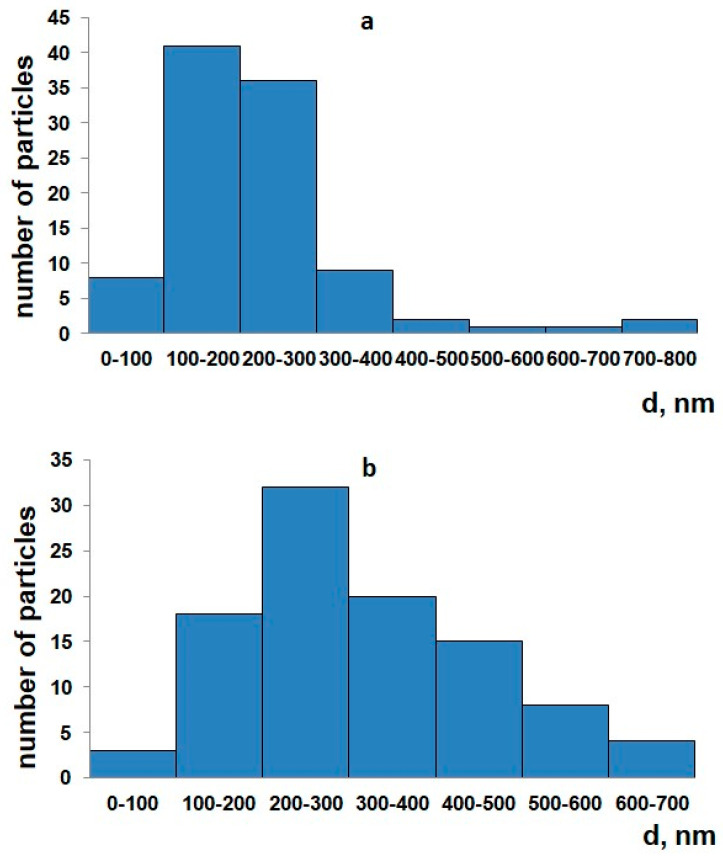
Histogram of particle size distribution for HAp (**a**) and CPs (**b**).

**Figure 8 ijms-25-02298-f008:**
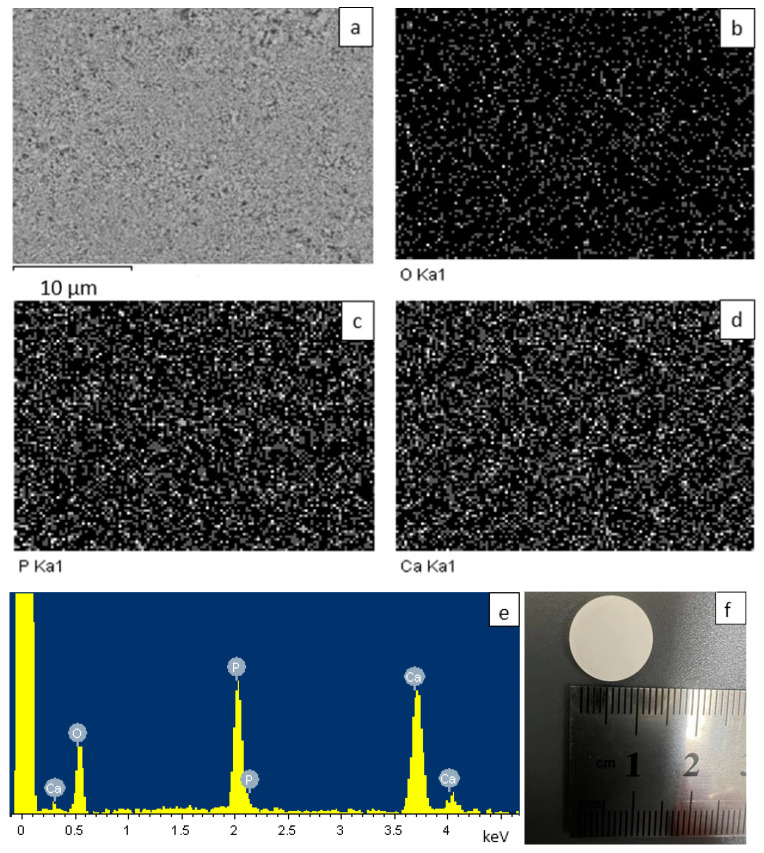
EDA analysis results: sample-scanning area (**a**); distribution of elements in the sample—O (**b**), P (**c**), and Ca (**d**); EDA spectrum (**e**); appearance of a compressed tablet (**f**).

**Figure 9 ijms-25-02298-f009:**
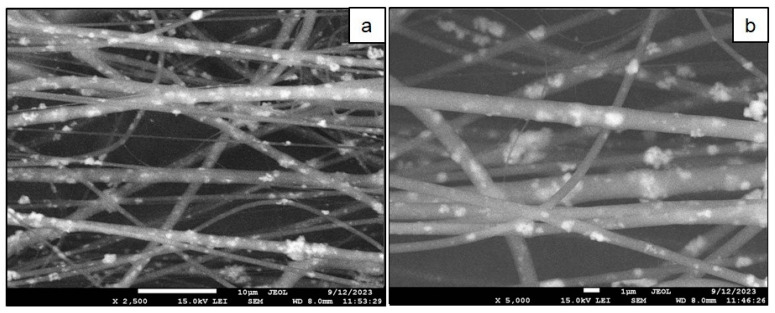
Photographs magnified 2500× (**a**) and 5000× (**b**) of the microstructures of the CPs fibers.

**Figure 10 ijms-25-02298-f010:**
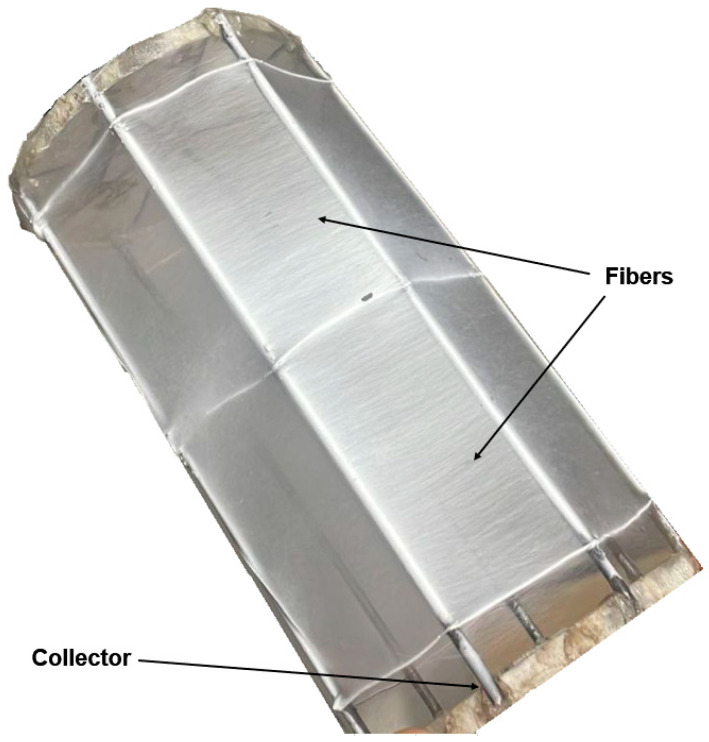
Photo of fibers transferred onto a collector.

**Figure 11 ijms-25-02298-f011:**
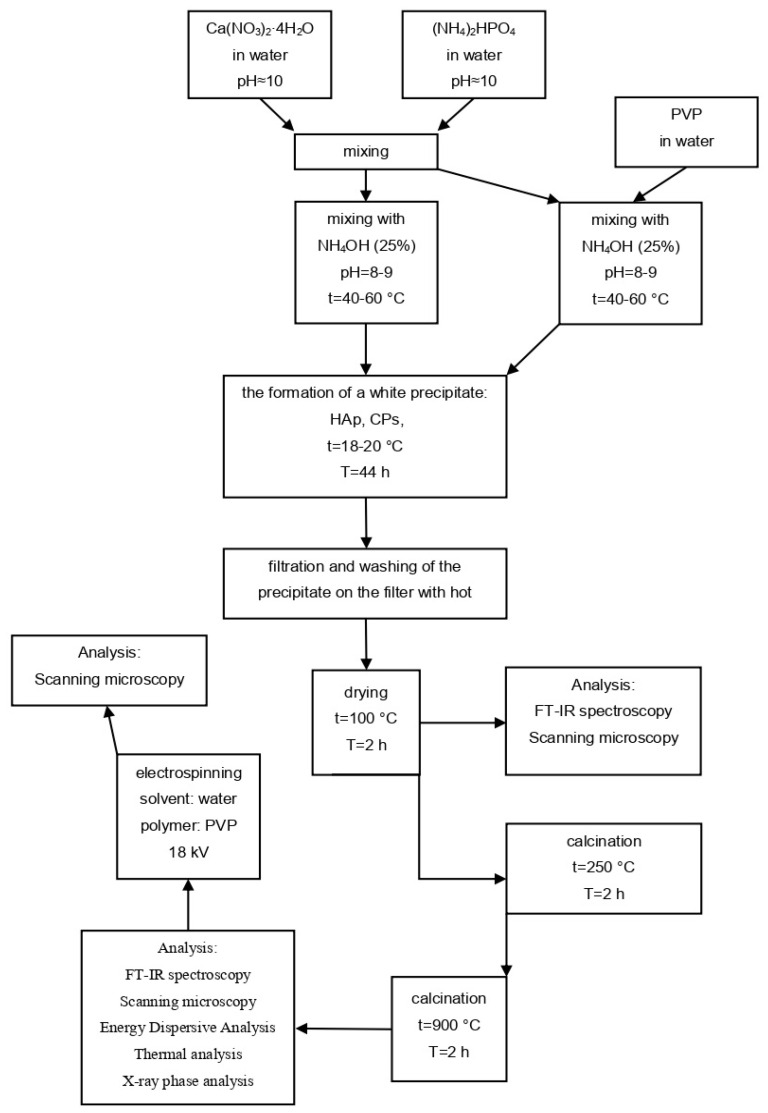
Flow chart depicting the steps of the study.

**Figure 12 ijms-25-02298-f012:**
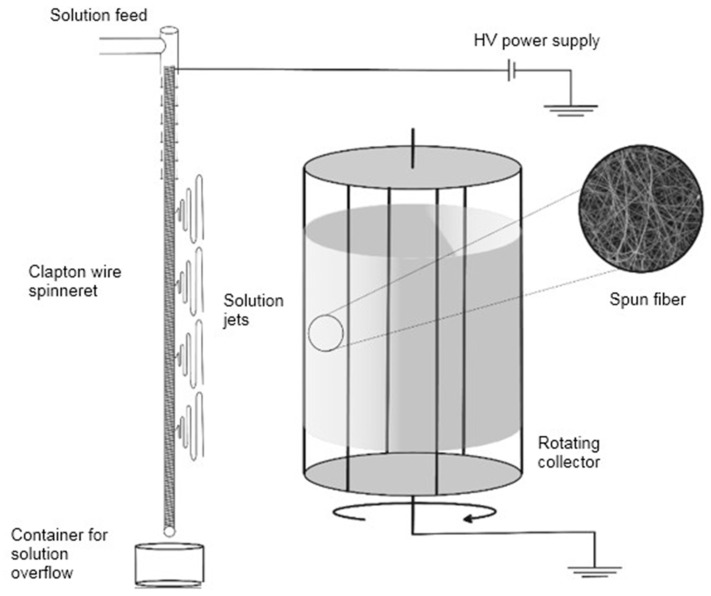
Scheme of needleless electrospinning setup.

**Table 1 ijms-25-02298-t001:** Lattice parameters of Ca_5_(PO_4_)_3_(OH), β-Ca_3_(PO_4_)_2_, and β-Ca_2_P_2_O_7_.

Chemical Formula	Crystal System	Space Group	a, Å	b, Å	c, Å	Alpha, °	Beta, °	Gamma, °
Ca_3_(PO_4_)_2_	Rhombohedral	R-3c	10.4290	10.4290	37.3800	90	90	120
Ca_2_P_2_O_7_	Tetragonal	P41	6.6840	6.6840	24.1450	90	90	90
Ca_5_(PO4)_3_(OH)	Hexagonal	P6_3_/m	9.4320	9.4320	6.8810	90	90	120

**Table 2 ijms-25-02298-t002:** Elemental composition (at%) of the synthesized samples.

	Samples	
	CPs	HAp
Ca	19.22	23.04
P	15.17	16.45
O	65.62	60.51
Ca/P	1.27	1.4

**Table 3 ijms-25-02298-t003:** The solubility of sample powders in physiological solution at pH 7, ω (NaCl) = 0.9%.

Samples	Concentration of Ca^2+^, mol/L	
	20 °C	37 °C
HAp	1·10^−3^	1.5·10^−3^
CPs	2.5·10^−4^	3.1·10^−4^
Fibers (CPs)	1·10^−4^	1.2·10^−4^

## Data Availability

Data and information related to this study are available upon request.

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
