# Peer review of "Obtaining Polyvinylpyrrolidone Fibers Using the Electroforming Method with the Inclusion of Microcrystalline High-Temperature Phosphates"

_ijms, 2024, doi:10.3390/ijms25042298_

Round 1

Reviewer 1 Report

Comments and Suggestions for Authors

The manuscript «Obtaining Polyvinyl Pyrrolidone Fibers by Electroforming Method with the Inclusion of Microcrystalline High temperature Phosphates» is well prepared for publication. I can give the following recommendations for corrections:

improve the quality of Figures 2 and 4 and increase the font size of the inscriptions on them by 1.5 times;

for EDS analysis, it is advisable to indicate on the SEM image the areas where the elemental composition was scanned;

it is not clear why the authors indicate the carbon content in Table 2, while samples 1 and 2 do not contain it (if the authors consider this important, it can be listed under the table with an asterisk);

the authors should double-check the elemental composition results, because according to X-ray diffraction data, samples 1 and 2 are swapped;

also, the materials and methods paragraph should come second, before the results.

Author Response

  1. «improve the quality of Figures 2 and 4 and increase the font size of the inscriptions on them by 1.5 times»

We have taken the reviewer's comment into account and enlarged the inscriptions on the figures accordingly.

  1. «for EDS analysis, it is advisable to indicate on the SEM image the areas where the elemental composition was scanned»

In the revised version of the article, we have included a photo of the compressed sample tablet's appearance, added a scanning area, and indicated the areas of element distribution over the surface.

  1. «it is not clear why the authors indicate the carbon content in Table 2, while samples 1 and 2 do not contain it (if the authors consider this important, it can be listed under the table with an asterisk)»

We believe that providing information on carbon content is important as it indicates that the polymer additive was completely burned out during annealing. In accordance with this comment, we have made additions to the text of the article.

  1. «the authors should double-check the elemental composition results, because according to X-ray diffraction data, samples 1 and 2 are swapped»

We appreciate this valuable comment and have made changes to the text of the article accordingly.

  1. «also, the materials and methods paragraph should come second, before the results»

Following the requirements for article formatting, we have placed the "materials and methods" section after the "discussion" section. The article now adheres to the structure of a magazine template, specifically IJMS.

Thank you for your attention and consideration.

Reviewer 2 Report

Comments and Suggestions for Authors

Manuscript ID: ijms-2857061

Dear Editor,

The authors discussed the synthesis mechanism of HAp and addition of PVP on the characteristics of HAp.

These issues should be clearly identified:

1-What is the Goodness of Fit (GOF) for the XRD reitvield analysis through using Highscore Plus software?

2-Instead of naming sample 1 and sample 2 throughout the text, it is better to label these compositions with more appropriate names.

3-the resolution of most figures are questionable. The authors should submit higher DPI figures.

4-The authors did not discuss the significance of obtaining these random fibers that incorporated nanoscale HAp inside on the mechanical, biological or any important properties associated with dental applications.

5-With the previous point, associated testing (either mechanical or biological) should be conducted.

Author Response

1 «What is the Goodness of Fit (GOF) for the XRD reitvield analysis through using Highscore Plus software?»

CPs consist of 34.5% β-Ca2P2O7 phase and 65.5% β-Ca3(PO4)2 phase. The Goodness of Fit (GOF) parameter for the HAp sample has a value of 1.67, and for the CPs sample, it is 1.79.

2 «Instead of naming sample 1 and sample 2 throughout the text, it is better to label these compositions with more appropriate names».

Thank you for your comment. We have made appropriate changes to the text of the article.

3 «Ð¢he resolution of most figures are questionable. The authors should submit higher DPI figures».

We have made appropriate changes and improved the quality of the figures.

4 «The authors did not discuss the significance of obtaining these random fibers that incorporated nanoscale HAp inside on the mechanical, biological or any important properties associated with dental applications».

We have taken into account the reviewer's comment and added data in the introduction and discussion of the results.

5 «With the previous point, associated testing (either mechanical or biological) should be conducted».

Experimental data were added to determine the solubility of a sample of CPs in fibers in a physiological solution at room temperature (20±1°C) and human body temperature (37±1°C).

Thank you for your attention and consideration.

Reviewer 3 Report

Comments and Suggestions for Authors

The manuscript requires major revisions for consideration for publication. Key areas for improvement include:

1. Introduction: Clarify the research problem and explicitly detail how your work addresses this issue.

2. Figure Quality: Enhance the clarity and professionalism of all figures, as their current quality is inadequate for presentation.

3. Figure Captions: Expand and detail the figure captions, using guidelines from scientific literature as a reference.

4. Results Section: Reorganize this section to present data coherently, with a clear narrative, rather than listing data points sequentially.

5. Discussion: Focus on interpreting the results, emphasizing their significance and the novel insights they provide. This should be done in the context of the addressed problem, with appropriate references to existing literature for a comprehensive perspective.

Comments on the Quality of English Language

 Minor editing of English language required

Author Response

  1. Introduction: Clarify the research problem and explicitly detail how your work addresses this issue.

Thank you for your comment. We have made appropriate changes to the "Introduction" section to clarify the research problem and explicitly explain how our work addresses this issue.

  1. Figure Quality: Enhance the clarity and professionalism of all figures, as their current quality is inadequate for presentation.

We have taken the reviewer's comment into consideration and improved the clarity and professionalism of all figures to ensure their suitability for presentation.

  1. Figure Captions: Expand and detail the figure captions, using guidelines from scientific literature as a reference.

In response to the reviewer's comment, we have expanded and detailed the figure captions, following guidelines from scientific literature as a reference.

  1. Results Section: Reorganize this section to present data coherently, with a clear narrative, rather than listing data points sequentially.

The results section has been reorganized to present the data coherently, with a clear narrative, rather than listing data points sequentially. This reorganization enhances the readability and understanding of the results.

  1. Discussion: Focus on interpreting the results, emphasizing their significance and the novel insights they provide. This should be done in the context of the addressed problem, with appropriate references to existing literature for a comprehensive perspective.

We have taken the reviewer's comment into account and made appropriate changes to the discussion section. The focus is now on interpreting the results, emphasizing their significance, and highlighting the novel insights they provide. This interpretation is done in the context of the addressed problem, with appropriate references to existing literature for a comprehensive perspective.

Thank you for your attention and consideration.

Round 2

Reviewer 2 Report

Comments and Suggestions for Authors

Required corrections were made.

Reviewer 3 Report

Comments and Suggestions for Authors

The authors have completed the necessary revisions to the manuscript. It is now ready for publication.